# Food Dishes for Sustainable Development: A Swedish Food Retail Perspective

**DOI:** 10.3390/foods10050932

**Published:** 2021-04-23

**Authors:** Linn Torstensson, Rebecca Johansson, Cecilia Mark-Herbert

**Affiliations:** Department of Forest Economics, Swedish University of Agricultural Sciences, P.O. Box 7060, 750 07 Uppsala, Sweden; linntorstensson@hotmail.se (L.T.); rebecca.johansson@gmail.com (R.J.)

**Keywords:** nutritional sustainability, portfolio management, sustainable diets, sustainable food consumption, sustainable nutrition

## Abstract

Current dietary patterns contribute negatively to greenhouse gas emissions and to the increased prevalence of non-communicable diseases. Earlier research on sustainable food consumption mainly focuses on diets, rather than single meals. Diets are difficult to measure, which is usually executed through self-reporting. This paper aims to identify frequently prepared dishes in a home setting through sales statistics, and how they can be altered to reduce climate impact and increase nutritional value. Commonly prepared food dishes in a home setting among customers of a major food retailer in Sweden were identified through sales statistics. The dishes were altered to reach nutritional and climate impact guidelines. Commonly prepared food dishes exceeded goals for climate boundaries by more than threefold and were not in line with nutritional guidelines. The content of fat, including saturated fat, was too high. Vegetables, fruits, wholegrains and fiber need to be increased. To reduce climate impact and increase nutritional value, the amount of animal-based product need to be reduced and/or exchanged to plant-based alternatives. This research contributes empirically to understandings of how portfolio management decisions influence food consumption based on sales statistics and how nutritional and climate impact guidelines can be applied from a single meal perspective.

## 1. Introduction

The largest cause of global environmental change is food production [1], which is responsible for up to 30% of global greenhouse-gas (GHG) emissions alone [2]. Climate changes are evolving fast, leading to devastating effects on the environment on a global scale [3]. Increasing incomes, rapid urbanization and inadequate accessibility to nutritious food are all factors in what has driven the global shift towards diets that are heavily processed, animal-based and high in calories [4]. Despite awareness of sustainable development challenges, the average Swedish diet also exceeds the global planetary boundaries for GHG emissions [3]. A shift toward environmentally sustainable diets is needed [5]. Swedish residents are consuming too much added sugar, saturated fat and sodium, and not enough fruit, vegetables, wholegrains or fibers [6]. These dietary transitions are not only responsible for contributing to environmental degradation [7]; they are also increasing the incidence of non-communicable diseases (NCD**s**) such as heart disease, cancer, Type 2 diabetes and stroke, causing premature deaths and lower global life expectancy [8]. NCDs are also contributing to major economic pressure in society [9].

Retailers have the potential to affect food choices through creating demand for specific products [10,11]. On a global level, multinational corporations stand for 69% of the 100 largest economies [12]. Therefore, they have a major responsibility towards society. Additionally, the consumer needs increased support from food retailers in order to enhance food choices with decreased climate impact [10] and higher nutritional value [13].

Previous work in the area of sustainable food consumption provides a broad background. The EAT-Lancet report [1], which is a full scientific review of what constitutes a healthy diet from a sustainable food system, provides a broad framework for dietary guidance. In addition to the EAT-Lancet report, Swedish governmental agencies (the Swedish Board of Agriculture in Jönköping, the Swedish Environmental Protection Agency in Stockholm and the Swedish Food Agency in Uppsala, together with the Institute for Food and Biotechnology) have set up a database to create a basis for discussion concerning climate factors and nutritional values in food [14]. Van Dooren et al. [15] framed diets with a low climate impact in line with nutritional national requirements. In their report, the researchers stated that it was possible to stay within the climate impact guidelines while obtaining an adequate nutritional diet. These reports did not take single meals into account, and the latter report had an overly generous climate budget. One Planet Plate is a guide to sustainable meals within a climate budget that also takes biodiversity into account [16]. However, so far, there are shortcomings in the frameworks and definitions for sustainable single meals that are also nutritionally adequate. It is also important to note that in order for people to undertake dietary changes, they need to be accepted from a socio-cultural perspective [17].

Dietary intake and patterns are challenging to measure [18]. Self-reporting is a common method of performing dietary assessment, but it can come with certain disadvantages such as inherent bias and recall bias. Indirect methods of dietary assessments, such as, for example, food balance sheets, are more appropriate for monitoring national dietary habits and nutrition patterns [19]. Furthermore, the perspective of food dishes provides an understanding of how and in what combinations foods are eaten [20]. These insights provide guidance on possible substitutions between different food groups are accepted from a behavioral perspective [20]. Therefore, this paper aimed to identify frequently prepared dishes in a home setting through sales statistics, and how they can be altered to reduce climate impact and increase nutritional value. The objectives were to: (I) identify the most commonly bought protein sources through sales statistics, (II) identify commonly prepared meals based around the protein sources, (III) calculate the nutritional content and climate impact of the food dishes and (IV) identify alternate dishes with higher nutritional value and lower climate impact. The Cynefin framework [21] served as an overarching conceptual theory for the analysis. It builds on the ontological framework of the Triple Bottom Line developed by John Elkington in 1999 [22].

To fulfill these objectives, the study is organized as follows. Section 2 describes the data and the unit of analysis, as well as the nutritional and climate framework. Section 3 presents the established dishes and their alterations. In Section 4, the results are discussed in the light of a plant-based transition and the integration of sustainable development in portfolio management. The final section summarizes the conclusions of the study, its contributions and suggestions for further research.

## 2. Materials and Methods

This chapter presents the research approach, including the research design, literature review procedure, data collection and quality control of the research process.

### 2.1. Case Study

This study was conducted through a case study research approach. A case study offers the possibility of examining a phenomenon in depth with a unit of analysis as a focal point [23]. Furthermore, it is an appropriate approach when there is a need to answer questions like “how” [24]. In this study, the aim was to identify commonly made dishes and explore “how” to transform them to alternatives with lower climate impact and higher nutritional value [25]. Therefore, a case study approach was chosen as a means to conduct this research.

### 2.2. Unit of Analysis—Meals Sold by Food Retail, Axfood

Swedish food retail is dominated by a few major actors [26]. Axfood represents one of the largest food retailers in Sweden and it was chosen as a unit of analysis. The criteria for selecting this unit of analysis related to proactive strategies for sustainable development, a willingness to share data and participation in the Sustainable Food Chain initiative [27].

### 2.3. Data Collection

Data were collected from Axfood’s receipt sales statistics from Enterprise Data Analytics (EDA) [28] and Axfood’s Consumer Insight Tool (ACIT) [29] based on data from all Hemköp and Willys food stores in 2019. Climate data were collected from the Mat.se climate database created in collaboration with Research Institutes of Sweden [30]. The climate data include information on certain foods and excludes data on factors after processing, for example, packaging, cooking and transport within Sweden [31]. These data are based on general assumptions on the climate impact of imported food commodities. The nutrition data originate from the Swedish food composition database [32] and use of the nutritional calculation program Dietist Net [33].

### 2.4. Data Analysis

Considering that protein sources tend to be the highest contributors to climate impact [34,35], the most sold food protein sources in weight were identified in the first step by sales statistics from EDA [28]. Foods commonly used as snacks, starters or spreads, e.g., sliced ham and caviar, were excluded. Eggs can be used in all types of dishes, which made it difficult to identify strong associations in the ACIT and were therefore excluded. In ACIT, associations were made based on purchase statistics, i.e., commodities consumers bought in combination with the protein source [29]. Through these associations, the researchers identified commonly purchased dishes. These dishes and the assessment of components were defined further by generic recipes from major databases such as ICA [36], Santa Maria [37], Köket [38] and Arla [39].

The nutritional content of the dishes was calculated per portion by their content of energy, protein, fat, saturated fat, wholegrains, fruit, vegetables and dietary fiber. The dishes were then compared and altered to be in line with the Nordic Nutrition Recommendations [40] suitable for average healthy adults. Additionally, carbon dioxide equivalents (CO_2_e) were calculated per portion for each dish through the Mat.se [30] climate database in order to be altered to the guidelines defined by the WWF [17]. In order to reach the climate impact boundaries and still have a nutritionally balanced meal with enough protein, it was not enough to reduce the amount of animal protein. Rather, it had to be fully replaced with plant-based protein alternatives, as they have a lower climate impact.

### 2.5. Nutritional Famework

The daily reference values for energy intake are based on the variety in gender, age and physical activity level [40]. Table 1 presents the dietary guidelines with recommended intakes per day and how these were applied per meal in this study.

The energy is divided into recommended energy percent (E%) among the macronutrients protein, carbohydrates and fat [38]. The right column in Table 1 shows the recommended intake per meal, which equals approximately 30% of the recommended daily intake [43]. However, no upper boundaries were set per meal on wholegrains, fiber, fruit and vegetables due to the remarkably low intake in the majority of Swedish consumers [7].

### 2.6. CO_2_ Boundaries

A scenario to reach the 1.5-degree UN target has been estimated by the WWF, based on the total use of CO_2_e per capita and how much that remains to be consumed by 2050 [17]. The guidelines based on this estimate for a meal within the planetary boundary concerning climate change are presented in Table 2.

To set a CO_2_e limit for a meal within the planetary boundaries, it was assumed that the allowance of total CO_2_e per capita per year would amount to 1.1 ton. The WWF concluded that 50% of the total 1.1 ton CO_2_e per capita would be allowed to come from food, with an increased proportion of the carbon budget for the food sector and a decrease in the other sectors, granted that all other sectors turn climate-neutral through carbon mitigation and production efficiency by the year 2050 [17]. This food budget would then account for 590 kg CO_2_e per year, 11 kg CO_2_e per week, 1.6 kg CO_2_e per day or 0.5 kg CO_2_e per meal (lunch or dinner).

### 2.7. Healthier Consumption Patterns with Lower Climate Impact

Consumption patterns are complex, as they consist of a variety of foods and dishes [40]. In order to transform consumption patterns to alternatives with a lower climate impact and higher nutritional value, several aspects have to be considered [3]. The diet needs to be based on minimal animal products with a varied range of minimally processed legumes, vegetables, fruits, wholegrains and tubers.

Garnett’s [3] 10 characteristics of a healthy diet with lower GHG emissions (illustrated in Figure 1) provided guidance for the analysis.

Figure 1 illustrates 10 different general principles of a healthier eating pattern with fewer GHG emissions. Although nutritional needs may vary in different population groups, these principles provide guidelines on how to construct more sustainable diets in both developed and developing countries [3].

## 3. Results

The empirical study is presented in terms of sales of protein as a starting point, leading the way to commonly prepared dishes, along with the climate and nutritional impacts of these meals.

### 3.1. Most Sold Protein Sources

The most sold protein sources by weight were identified through sales statistics, displayed in Figure 2.

The most sold protein sources were, in falling order: chicken breast fillet, minced meat (beef), chicken leg, grilled or spicy sausage, minced meat (mixed pork and beef 50/50), Falu sausage, pork fillet, salmon fillet, bacon and whole chicken. Chicken was in the largest percentage category and present in three different categories. In total, chicken represented 38% of the top 10 sales. Red and processed meat together made up 56%, i.e., the majority. Fish represented 6% of the top 10 protein sources sales.

### 3.2. Commonly Prepared Dishes

Associations in sales statistics were used to define commonly prepared dishes, presented in Table 3.

The 10 most sold protein sources are presented in the left column, followed by commonly prepared dishes for each protein source in the right column. Some associations indicated several dishes being made with the same protein source. On the contrary, some of the protein sources had weak associations, which made it difficult to identify commonly made dishes. Therefore, bacon and whole chicken were replaced with two dishes that were based on chicken breast fillet and minced meat.

### 3.3. Alteration of the Food Dishes

Several ingredients needed to be changed for the dishes to be in line with nutritional and climate impact guidelines. The changes are presented in Table 4.

The left column presents the considered guidelines aspects, followed by the main changes made in the original dishes for each of them to be in line with the recommendations stated in Table 1 and Table 2.

In order to reduce total fat content, including saturated fat, the amount of animal-based product was reduced or, to some extent, exchanged with low-fat or plant-based alternatives such as low-fat dairy products, soy-based mince and oat-based cream. The exchange and reduction of animal-based products were also made to reduce the amount of CO_2_e. Seafood with a high climate impact such as salmon and shrimp were replaced with lower climate impact seafood such as mussels and clams. Furthermore, wholegrain products such as wholegrain couscous were added to increase the content of wholegrains and dietary fiber. The content of dietary fiber was also increased through added fruits and vegetables.

### 3.4. Nutritional Content and Climate Impact of the Food Dishes

In order for the dishes to be in line with the nutritional and climate guidelines, alterations were made. Figure 3 presents the average nutritional content and climate impact of the original dishes and the altered dishes in comparison with the guidelines. The values are defined in percent compared with the guidelines.

In Figure 3, the nutritional aspects and climate impact (carbon dioxide equivalents) are listed to the left. The upper line represents guidelines’ recommendations, followed by the original dishes and the altered dishes. In the original dishes, the content of energy, protein, fat, saturated fatty acids and climate impact exceeded the guidelines. Energy and protein content were within the range of recommended intake but above average. The content of fat and saturated fat and climate impact were especially high and exceeded the guidelines at 66% (44.1 g), 96% (17.7 g) and 236% (1.68 kg CO_2_e) respectively. On the contrary, wholegrains, dietary fiber, fruits and vegetables were significantly lower in the original dishes compared with the guidelines. None of the original dishes contained wholegrains. The content of dietary fiber, fruits and vegetables was 44% (4.2 g) and 59% (60.9 g) below the minimum requirements.

The altered dishes were in line with nutritional and climate impact guidelines, although the content of wholegrains, dietary fiber, fruits and vegetables exceeded the minimum requirements by 115% (45.1 g), 132% (17.4 g) and 31% (196.7 g). The content of saturated fatty acids was 43% (5.1 g) below the maximum level presented in Table 1.

## 4. Discussion

In this section, the empirical results are discussed in relation to different protein choices, a plant-based transition, climate impact, nutritional values and the integration of sustainable development in portfolio management.

### 4.1. Consequences of the Choice of Protein Source

The results show that the major protein sources among the customers of the second biggest food retailer in Sweden seem to be meat, which supports the results reported by Amcoff et al. [7]. According to the Nordic Council of Ministers [40], a Western diet typically includes high amounts of red meats. However, for health reasons, it is preferable to eat a variety of fruit, vegetables and legumes. Earlier research has indicated that a vegetarian diet is related to a decreased risk of NCDs such as obesity, Type 2 diabetes and cardiovascular diseases [45,46,47]. Furthermore, consumption of red meat increases the risk of colorectal cancer [48]. On the other hand, the exclusion of animal-based protein sources in the diet can lead to inadequate or deficient levels of certain nutrients [40]. Plant-based proteins such as legumes, seeds, nuts and wholegrain cereals have lower protein quality due to the imbalance of essential amino acids. Animal protein sources have a better balance between essential amino acids and, therefore, high protein quality. Even so, it is possible to obtain a plant-based diet with good protein quality by including a variety of vegetarian protein sources in the diet. Other important nutrients to take into consideration are Vitamin B12 and iron. Vitamin B12 only exists in animal-based foods and, therefore, needs to be supplemented or obtained through fortified foods in a plant-based diet. Plant-based food, for example nuts, legumes and wholegrains, has a comparatively high content of iron. However, the type of iron differs from that of animal-based foods. Animal-based foods contain heme iron, which has higher bioavailability compared with the non-heme iron present in plant-based foods. It is possible to facilitate the absorption of non-heme iron in plant-based alternatives if combined with foods rich in vitamin C [40].

Animal-based protein is problematic from a climate change perspective, but if other sustainability factors are included, both socio-economic and environmental, these complicate the choices for sustainable development [49]. Animal products have a high climate impact in comparison with plant-based alternatives [35,50]. According to the results, chicken was the most commonly bought protein source. Even though chicken meat contributes to fewer GHG emissions than other types of meat, the contribution is still substantially higher compared with plant-based protein sources [34]. Moberg et al. [5] stated that animal products are responsible for more than half (about 67%) of the climate impact caused by the average diet in Sweden. Therefore, it would be preferable for both public health and the environment to change the consumption to more plant-based protein.

### 4.2. Commonly Prepared Dishes

It is urgent to increase awareness of the environmental effects of animal-based protein [21]. Additionally, food choices are complex and affected by several factors, including lifestyle, values, personal needs and preference, sensory and perceptual factors, habitual patterns, available resources and knowledge, social factors and the physical environment [51,52]. De Boer and Aiking [21] stated that scientific-based arguments may not be enough for consumers to adapt to more sustainable food choices. In this context, food dishes might facilitate the transition. According to Meiselman et al. [53], consumers commonly exhibit food acceptance through assimilation, meaning that it is more likely that they will enjoy a certain dish if the expectations are already high. Further acceptance of dishes with less meat and more plant-based options are facilitated by the type of recipe and how well presented the dishes are [54]. Lastly, spiciness could be an auspicious strategy for increased acceptance of plant-based dishes, as these dishes have shown to be equally approved as its meat counterpart. With regards to Spencer et al. [54], research has demonstrated that a substitution from meat-based dishes to plant-based alternatives is possible in different populations. Furthermore, the statistics show that meat consumption is decreasing among Swedish consumers [55]. These results indicate that the dishes in this study are likely to be accepted. Several of the compiled food dishes in this study are similar to self-reported dishes from Food and Friends [56]. Food and Friends is a trendspotting report where 1000 Swedish people are asked annually to self-report what and how they eat. These dishes were spaghetti Bolognese, tacos, sausage stroganoff and dishes with salmon/fish, pork fillet, chicken and pasta. According to Food and Friends [56], dishes with chicken and spaghetti Bolognese were the most frequently consumed foods both on weekdays and during the weekend. In this study, chicken and minced beef were commonly purchased protein sources. However, there are some differences between the compiled dishes in this study and the dishes in Food and Friends [56]. The differences could depend on a variety of aspects, where one aspect could be that this study limited the number of dishes to 10, whereas Food and Friends included 20 dishes. Food and Friends also included eating out of home settings, which is not visible in sales statistics from food retail stores and presents different types of dishes, such as pizza and hamburger. It could also depend on different types of dietary assessment method. Food and Friends is dependent on self-reporting, where recall bias may be present, and respondents can be selective with the foods they choose to report or have a hard time estimating proper food intake [7,20].

In this study, associations were made to define the dishes based on sales statistics though chosen generic recipes, where the authors’ own food habits could have affected the outcome of the defined dishes. Furthermore, the sales data do not foretell what food products the consumers already have at home or buy at other stores or what is actually consumed or wasted. Besides this, sales do not reflect total consumption, as there are losses along the food chain and products which are not sold through food stores. Dietary consumption can be difficult to measure, and no single method assesses dietary intake perfectly [19]. The consumer purchases of this study are based on all purchases made in Willys and Hemköp during one year. Hence, the data collected in this study provide a larger basis without risk of self-reporting bias. However, these data do not capture purchases for consumers in Sweden as a population, but rather the segment that purchases food in food stores owned by Axfood. Nonetheless, combining self-reporting with sales statistics gives a more accurate estimate of dietary intake than that of individual methods. Hence, this study confirms that it is most likely to be common to cook dishes with chicken and minced beef in a home setting.

### 4.3. Climate Impact and Nutritional Value of Commonly Prepared Dishes

All dishes exceeded the boundaries set for the study on the maximum 0.5 CO_2_e per portion. The average CO_2_e for all 10 dishes was more than threefold the recommended boundary, at 1.68 CO_2_e per portion. It is important to take into consideration that the CO_2_e values should not be interpreted as absolute numbers but rather as guidelines [34]. Dishes containing beef had the highest climate impact, and this is because of enteric fermentation in ruminants causing the potent GHG methane to be released [57]. Dishes containing chicken had the lowest climate impact, and that might be due to the low feed conversion ratio with high protein retention [58]. The other dishes ended up between beef and chicken. These results support the findings by Röös [34] that reported on beef having the highest impact, followed by pork and chicken as the lowest. Recently conducted research confirmed that the Swedish diet exceeds the planetary boundaries by more than two to three times [5]. This study indicates that the dishes exceed the climate boundaries, which is one of the planetary boundaries, by more than threefold. It is probable that this might be higher, due to only lunches and/or dinners being included in this study, while Moberg et al. [5] included the whole diet. Breakfast and snacks usually do not contain the same amount of animal protein, which could lower the overall climate impact.

It is problematic that the selected dishes were inadequate for the nutritional recommendations due to the fact that healthy food habits can reduce the increased prevalence of non-communicable diseases [40]. However, energy balance was not generally exceeded in the dishes, which was unexpected, considering the increased prevalence of obesity [59]. This result may come from the portion size boundaries within this study, having no regard for the actual sizes of meals consumed in home settings. Compared with Riksmaten [7], a national survey of Swedish adults, this project also showed meals with insufficient content of fruits, vegetables, wholegrains and dietary fiber. They differed in total fat content, where this study showed an exceeded amount of fat while Riksmaten presented the recommended amount of fat intake. Both showed excessive amounts of saturated fat intake. Compared with this study, Riksmaten includeed all foods consumed as a means to define dietary patterns. Potentially, the total fat intake in Riksmaten was balanced by including all meals consumed, instead of only specific dishes, as done in this study. Furthermore, the dietary assessment methods differed between this study and Riksmaten, where the latter was a self-reporting survey. This could also have possible implications for overall nutritional value due to different types of bias [20]. Nonetheless, both Riksmaten and this study concluded that the average food intake in Sweden generally contains too few fruits, vegetables and wholegrains; too little dietary fiber and too much saturated fat.

### 4.4. Possibilities to Reduce Climate Impact and Increase Nutritional Value

The climate impact of the food dishes was mainly reduced by reduction and/or exchange of animal products to plant-based alternatives. It confirms earlier research arguing that the amount of meat and animal products offers the greatest potential to reduce climate impact [60]. In order to reach the target of 0.5 CO_2_e per dish and portion, the amount of animal products had to be altered.

Willet et al. [2] composed a diet within planetary boundaries based on mainly vegetables, fruits, nuts, legumes, wholegrains and unsaturated oils, with small amounts of seafood. Although Willet et al. [2] included a wider range of sustainability aspects, the content and proportion of the diet was similar to the altered dishes in this study. The main divergence is the content of red meat, starchy vegetables and refined grains, which were excluded or only allowed in limited amounts in the diet suggested by Willet et al. [2]. In this study, poultry and red meat were totally excluded in the altered dishes. A possible explanation for this is that the dishes in this study were handled individually rather than from a diet perspective. Hence, the nutritional and climate impact guidelines were rigid and did not allow broad ranges. If the dishes had been handled from the perspective of a weekly diet, like the one presented in One Planet Plate [17], broader ranges in nutritional value and climate impact would have been manageable. Furthermore, the altered dishes aimed to be similar to commonly made dishes, which made the framework for alteration even more rigid. From a climate perspective, it would have been possible to allow limited amounts of poultry in some of the altered dishes. Food choices are highly affected by habitual patterns shaped by earlier reflection and food choices [51]. If poultry were to be included, the amounts would have been significantly reduced to still be within the climate impact boundaries. Hence, the dishes would entail other proportions, which could possibly create a dish further away from habitual patterns.

To increase the nutritional value of the dishes presented in the current study, wholegrains, fruits and vegetables were added in accordance with the Nordic Nutritional Recommendations [40]. The content of fat, including saturated fat, was reduced and altered through replacement with unsaturated fatty acids from plant-based alternatives. Red meat was also replaced with a plant-based alternative, which contributed further to the reduced content of saturated fat.

### 4.5. Integration of Sustainable Development in Portfolio Management

Every corporation’s path to sustainability differs and it needs to be customized in accordance with the specific organizational objectives at all levels [61]. When sustainability is incorporated, it offers the possibility to create value and competitive advantage. Limited research has involved how to integrate sustainable development in portfolio management [62,63]. Green product portfolio decisions and strategic sustainability decisions offer some possibilities [60]. Sustainable products could represent larger shares in the product mix and increased marketing efforts, which would decrease the share of less sustainable alternatives. Development of new or existing products provides further possibilities [60]. Hence, the dishes could offer possibilities for future product development, campaigns and consumer-oriented information. Since food dishes are connected to practical habits in home food preparation, it is important that the developed food products can be used and prepared in the same context [21]. In conclusion, the expansion of plant-based protein alternatives may be part of the solution to more sustainable diets.

A challenge to this is that it requires changes in the strategic management of the retailers [12]. Sales growth defines the power dynamics of sector structure. Therefore, it is a fundamental part of the growth strategies of the retailers, which also increases their influence over consumers and their food choices. Currently, retailers are shaping dishes by linking product promotions that provide items for a complete meal and branding their own ready-made meal items [12]. Tjärnemo and Södahl [64] stated that the meat category is vital for Swedish food retailers from a financial perspective. Overall, the meat category represents a large share of total sales. Additionally, it is an important section in the food store that affects the customer’s choice of retailer [64]. The incentive for the retailers to change their sales policy in this context is that the awareness of sustainability is growing among the consumers [65]. Consumers are expecting corporations to act and “walk the talk”. Businesses today experience intense scrutiny, where lack of awareness or transparency in issues regarding sustainability might damage the brand [66]. Hence, integrating sustainability provides possibilities for improved consumer value and a stronger corporation, resulting in increased economic value [65].

## 5. Conclusions

Human diets need to change in order to decrease the growing prevalence of non-communicable diseases and halt the devastating consequences of climate change. Since food choices are highly affected by habits, the food dishes analyzed were the most commonly prepared dishes in a home setting, based on sales statistics from one of the largest food retailers in Sweden. All dishes identified in this study exceeded the goals for climate impact and were nutritionally imbalanced. They were all lacking in fiber, fruits, vegetables and wholegrains while consisting of too much total and saturated fat. Therefore, fiber, fruits, vegetables and wholegrains were added in the altered dishes to reach the nutritional and climate guidelines for each dish. Furthermore, animal products were reduced or traded for plant-based alternatives.

Increased knowledge is needed on how to continue the integration of sustainability aspects in single-meal food consumption. This project points to possibilities for food retail portfolio strategies to gradually enable transitions of food consumption habits through product offers and services. Social acceptance may be created through, for example, nudging strategies [67] and transparency to make it easy for the consumer to understand information at a product level. Further knowledge is needed on how to implement and guide the consumer to more sustainable alternatives. More research is also needed on how to apply this kind of knowledge in different sectors of food systems to facilitate the required transformation to more sustainable food systems.

This research contributes to confirming earlier conducted studies on food consumption in Sweden through a different dietary assessment method based on sales statistics rather than self-reporting. Food retailers possess valuable data which can contribute to the research field and thereby contribute to a more sustainable future. Therefore, it is important that food retailers are more transparent and willing to share their information. Furthermore, the study shows how nutritional and climate impact guidelines can be integrated to create more sustainable food dishes in everyday life as a contribution to more sustainable food consumption overall. From a sustainability perspective, it would be valuable to further examine the possibility to combine different aspects related to the environmental, social and economic aspects of sustainability.

One of the limitations of this study is that the data are based on one of the three big retailer corporations in Sweden (Axfood). For further research, it would also be interesting to execute similar analyses in other Swedish food retailers to compare sales statistics and other data for a broader perspective on national food consumption. In order for the results to contribute to more sustainable food systems, this needs to be utilized. Therefore, further research could examine how to integrate this kind of information in practice within the food retail industry and other sectors as a means to reach the sustainable development goals.

The limited number of studies related to fast-moving consumer goods portfolio management for sustainable development in the food sector is surprising. There ought to be an abundance of studies in the field, due to food production being the largest cause of global environmental change and because all consumers, in fact, are dependent on food for sustenance. However, portfolio management can be viewed as an internal business matter and not something that is willingly shared or researched, and this outlook may be the cause of the shortage of related studies. Nonetheless, this type of management only benefits short-term profits for the corporation, but does little for corporate responsibility, transparency and efforts to improve public health.

## Figures and Tables

**Figure 1 foods-10-00932-f001:**
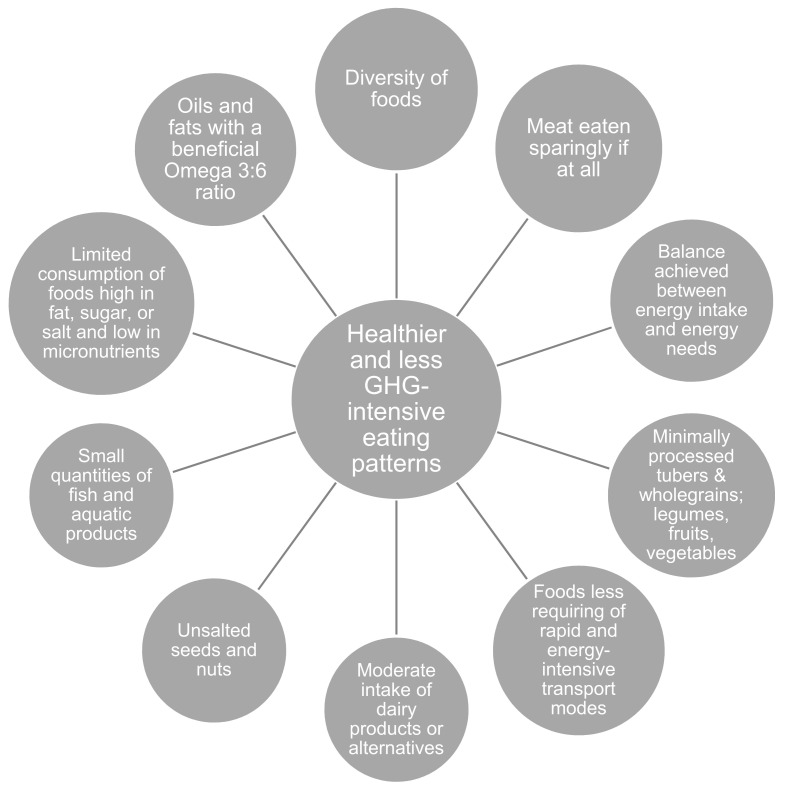
Characteristics of healthier and less land-intensive eating patterns with fewer greenhouse gas emissions (own version according to Garnett [3]).

**Figure 2 foods-10-00932-f002:**
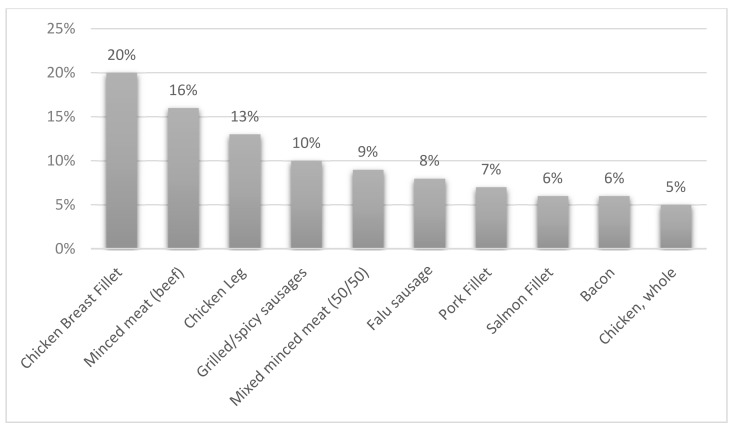
The 10 most sold protein sources at all Hemköp and Willys stores during 2019 and their relative percentage (of 100%). Data gathered from Axfood’s sales statistics Enterprise Data Analytics (EDA) [28].

**Figure 3 foods-10-00932-f003:**
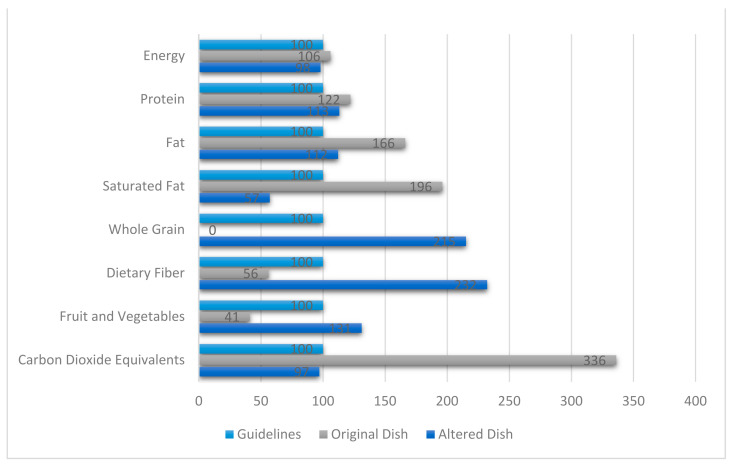
Average nutritional content and climate impact (kilogram CO_2_e) of the original and altered food dishes per portion in comparison with the Nordic Nutrition Recommendations [40], the Swedish Food Agency [41,42] and the climate impact goals set by WWF [44]. The x-axis values are defined in percent relative to the guidelines presented in Table 1 and Table 2.

**Table 1 foods-10-00932-t001:** Dietary guidelines for a healthy diet, gathered from the Swedish Food Agency [41,42] and the Nordic Council of Ministers [40], with reference values for energy and nutrient content in an average school lunch, corresponding to 30% of the recommended daily intake [43].

Component	Recommended Intake Per Day (100%)	Recommended Intake Per Meal (30%)
Energy (kcal)	1700–3200 kcal	735 kcal (510–960)
Protein	10–20 E% *	18–37 g
Fat	25–40 E% *	20–33 g
Saturated fatty acids	<10 E% *	<9 g
Wholegrains	70–90 g **	>21 g
Dietary fiber	25–35 g *	>7.5 g
Fruits and vegetables	500 g *	>150 g

* Ref. [40]; ** Refs. [41,42].

**Table 2 foods-10-00932-t002:** Reference values for CO_2_e recommendations per year, week, day and meal, the latter corresponding to 30% of the recommended daily intake [44]. WWF: World Wildlife Fund.

Organization	CO_2_e Recommendation per Year	CO_2_e Recommendation per Week	CO_2_e Recommendation per Day	CO_2_e Recommendation per Meal (30%)
WWF	590 kg	11 kg	1.6 kg	0.5 kg

**Table 3 foods-10-00932-t003:** The 10 food dishes based on consumer purchase associations from ACIT * for each protein.

Protein Source	Food Dish
Chicken breast fillet	Tikka masala with rice, chicken wok with noodles
Minced meat (beef)	Taco, spaghetti Bolognese
Chicken leg	Chicken leg with curry sauce and rice
Mixed minced meat (50% pork, 50% beef)	Lasagna
Falu sausage	Sausage stroganoff
Pork fillet	Marinated pork fillet with pre-cooked potato wedges and Béarnaise
Salmon fillet	Salmon and shrimp with lemon and dill sauce and boiled potatoes

***** Axfood’s Consumer Insight Tool [29].

**Table 4 foods-10-00932-t004:** The main ingredient changes made from the original dish to the altered one to shape the meals in accordance with the Nordic Nutrition Recommendations [40] and climate impact goals set by the WWF [44].

Guideline Aspect	Problems with the Original Dish	Altered Dish
Total fat content and saturated fatty acids	Too high, mainly from red meat and animal-based dairy products with high fat content	Animal-based products with lower fat content combined with or changed to plant-based alternatives
Wholegrains	None	Added wholegrain alternatives
Dietary fiber	Not enough dietary fiber, small amount of fiber through sparse vegetables	Added wholegrain alternatives, increased amounts of fruits and vegetables
Fruit and vegetables	Not enough included	Increased amounts of fruits and vegetables that are less energy-intensive
Carbon dioxide equivalents	Too high, mainly from meat and other animal-based products	To a large extent, substituted with plant-based protein alternatives and plant-based dairy alternatives

## Data Availability

Confidential sales data supporting the reported results can be found within Axfood’s receipt sales statistics Enterprise Data Analytics (EDA) and Axfood’s Consumer Insight Tool (ACIT). Climate data were gathered from Mat.se, available at https://www.mat.se/mat-klimat, where different food products are labeled with CO_2_ equivalents. Nutritional data are available at the Swedish Food Agency’s database at http://www7.slv.se/SokNaringsinnehall/.

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
