# Peer review of "Food Dishes for Sustainable Development: A Swedish Food Retail Perspective"

_foods, 2021, doi:10.3390/foods10050932_

Round 1
Reviewer 1 Report
Interesting article, which could be improved with the following suggestions.
1. The authors should explain or indicate under which theoretical approach the work is based (indicate it in the introduction).
2. For the reading to be more fluid, the authors should include the common thread of the work at the end of the introduction.
3. The term “sustainable development” appears in the title of the work. I think the authors could expand the theoretical framework by referring to this concept. In this sense, it would be interesting to analyze this article from the Triple Base Line theory in which the three main dimensions of sustainability are defined (economic sustainability, social sustainability and environmental sustainability), and point out under what dimension and dimensions of this theory the work could be sustained.
Thank you.
Reviewer 2 Report
Results of this case study illustrate the potential for creating climate friendly meals by modifying traditional recipes using ingredients that contribute less to global greenhouse gas emissions. From my perspective, the study was well-designed, and all sections of the manuscript are clear and thorough. The Discussion and Conclusions address important considerations, implications, and applications of the results. I have only a few thoughts/questions and suggestions.
Please clarify in methods that poultry and red meat were totally excluded, as indicated in Lines 419-420 of the Discussion. Table 4 suggests to me that some of these animal products were reduced in amount. For “Total fat and saturated fat”, it says combined or exchanged, and for “Carbon dioxide equivalents” it says, “To some extent substituted”. It would be helpful to provide some concrete examples of the modifications or show an example of one or two recipes with alterations.
How was the salmon and shrimp dish modified?
Lines 334-335: Is there any data on attitudes/sales/ consumption of plant-based protein alternatives in Sweden? The results discussed in this paragraph do not necessarily indicate to me that dishes in this study are likely to be accepted, only that there is potential for modification and acceptance using different strategies.
Line 419: I suggest replacing “In this study” with “In the current study….”
Lines 430-432: Just a thought, would the plant-based alternatives to poultry be more acceptable than modifying a dish to include less poultry and higher proportions of other ingredients?
Reviewer 3 Report
This is an interesting research to explore food nutrition and sustainable development. The paper is well written with in-depth analysis and discussion. Industry dataset was applied to reach reliable assumptions and recommendations. I would like to raise several minor issues:
- Lines 33-34, there is a logic jump here which mentioned Sweden, global and then Sweden, suggest to alter the sentence sequence; also the reference for this sentence is needed;
- Figure 1, the fonts should be smaller to have the phrases inside the circles;
- Line 279, change “In this chapter” into “In this section”;
- A few limitations can be provided in the conclusion such as limited data size, Sweden background.
